# Pulmonary Thrombosis despite Therapeutic Anticoagulation in COVID-19 Pneumonia: A Case Report and Literature Review

**DOI:** 10.3390/v15071535

**Published:** 2023-07-12

**Authors:** Cristian-Mihail Niculae, Maria-Evelina Gorea, Laura-Georgiana Tirlescu, Rares-Alexandru Constantin, Ruxandra Moroti, Adriana Hristea

**Affiliations:** 1Infectious Diseases Department, Faculty of Medicine, “Carol Davila” University of Medicine and Pharmacy, 37 Dionisie Lupu Street, 050474 Bucharest, Romania; cristian.niculae@drd.umfcd.ro (C.-M.N.); ruxandra.moroti@umfcd.ro (R.M.); 2National Institute for Infectious Diseases “Prof. Dr. Matei Bals”, 1 Calistrat Grozovici Street, 021105 Bucharest, Romania; maria-evelina.gorea@rez.umfcd.ro (M.-E.G.); laura-georgiana.tirlescu@rez.umfcd.ro (L.-G.T.); rares-alexandru.constantin@rez.umfcd.ro (R.-A.C.)

**Keywords:** COVID-19 pneumonia, inflammation, pulmonary thrombosis, risk factors, anticoagulation

## Abstract

The rate of thrombotic complications in COVID-19 patients is high and could be associated with the risk of unfavourable outcomes. Moreover, pulmonary thrombotic events can occur even in patients already on anticoagulant treatment. We present the case of a patient with severe COVID-19 pneumonia, without traditional risk factors for thrombosis, who developed massive pulmonary thrombosis (PT) despite therapeutic anticoagulation. The diagnosis was challenging, and the case raised concerns about the protective role of conventional anticoagulant treatment in COVID-19 pneumonia. Thus, we searched for literature reports on COVID-19 patients who developed PT despite being under anticoagulation therapy. We identified 13 cohort studies including 4058 patients of which 346 (8.5%) developed PT and nine case reports/series enrolling 14 patients. Four cohorts were further analysed, which reported data on risk factors for thrombosis, outcomes and biological characteristics. We found that there were no differences between patients with and without PT regarding the classical risk factors for thrombosis. PT occurred regardless of the anticoagulation regimen, and the risk factor identified was severe COVID-19 pneumonia and a stay in an intensive care unit (ICU). Pulmonary thrombotic events in patients with COVID-19 are rather inflammation-related than correlated with traditional thromboembolic risk factors, and the therapeutic approach must take into consideration this aspect.

## 1. Introduction

Coronavirus disease 2019 (COVID-19) is a complex respiratory and systemic disease that can progress to severe inflammation, which can lead to a prothrombotic state [1,2,3,4]. The most common vascular thrombotic complication involves the pulmonary arteries, mostly as a peripheral in situ microthrombosis that can be found in areas with associated inflammatory lung lesions [2,5,6,7,8,9,10]. The rate of thrombotic complications in COVID-19 cases has been reported to occur in up to 35% of patients, being associated with the risk of unfavourable outcomes [2]. Adequate screening, diagnostic and antithrombotic strategies are essential, as there is still debate regarding the optimal clinical approach for these patients [11,12,13].

The failure of anticoagulation treatment in thrombotic diseases is recognised as a potential problem, especially among patients with coexistent active cancer [14,15]. In the general population, 5–10% of all acute ischemic strokes occur despite anticoagulation in patients with nonvalvular atrial fibrillation because of multiple causes like subtherapeutic doses or noncompliance to the treatment, malignancy or atherosclerotic pathology [16]. In some patients under direct oral anticoagulants (DOACs), even if adequately anticoagulated, the occurrence of thromboembolism (whether primary or recurrent) constitutes treatment failure, which has been reported in approximately 2% of patients in large DOAC clinical trials [17]. The lack of adequate monitoring could be an issue for these patients [17].

Pulmonary thrombotic events can occur in patients with SARS-CoV-2 pneumonia despite thromboprophylaxis [4,18,19,20,21]. Nonpulmonary major arterial thrombotic complications have also been reported in severe and critical COVID-19 cases, despite oral or low-molecular-weight heparin (LMWH) anticoagulation, with high mortality rates [16,22]. 

We present a clinical case reflecting the problems in differential diagnoses and the protective role of conventional anticoagulant treatment to prevent pulmonary thrombotic disease in a patient with COVID-19 pneumonia and no other risk factors for thrombosis. We also conducted a literature review of similar reports to provide useful clinical and anticoagulation data for preventing pulmonary thrombotic events in patients with SARS-CoV-2 infection.

## 2. Case Report

A 59-year-old woman was admitted to our hospital in January 2021 with symptoms of fever, cough and shortness of breath for eight days and a positive SARS-CoV-2 RT-PCR nasopharyngeal swab. The patient had no significant past medical history and did not take any medications. She was unvaccinated against COVID-19. The clinical examination showed acute respiratory failure with respiratory distress: a respiratory rate of 26/min and a need for supplemental oxygen therapy by face mask of 15 L/min to maintain normal peripheral oxygen saturation. The blood pressure (BP) was 145/85 mmHg, and the heart rate was 94 beats/min. Other systemic examinations were unremarkable. Blood samples revealed lymphopenia (600 cells/mm^3^) and high values of LDH (827 U/L) and inflammatory markers (C-reactive protein of 145 mg/L, fibrinogen = 440 mg/dL, ferritin > 1650 ng/mL, IL-6 = 60 pg/mL). Normal values were recorded for D-dimers (338 ng/mL), platelet count (PLT), cardiac markers and liver and renal function. The ECG was unremarkable. Severe COVID-19 interstitial pneumonia with 50% lung involvement was described on the chest CT scan. 

She received supplemental oxygen and therapeutic anticoagulation with LMWH, dalteparin 5000 UI s.c. at 12 h from the admission, antiviral treatment with Remdesivir 200 mg i.v. loading dose (first day), then 100 mg i.v. qd for the next six days, dexamethasone 8 mg i.v. qd and immunomodulatory treatment with anti-IL-1, Anakinra 100 mg s.c. bid (first three days), then 100 mg s.c. qd. The initial evolution of the patient was slightly favourable, in terms of respiratory status and inflammatory markers. 

On the seventh day after hospital admission, the patient developed a sudden deterioration of respiratory function, with a raised body temperature of 37.6 °C and low BP (88/55 mmHg) requiring intensive care unit (ICU) admission, noninvasive mechanical ventilation (NI-MV) and cardiovascular support. The arterial blood gases showed marked hypoxemia (PaO_2_ = 39 mmHg, SaO_2_ = 70%) and a high lactate value (7.6 mmol/L). The blood sample analysis was as follows: marked leukocytosis with neutrophilia (30,160 cells/mm^3^); increased levels of CRP (83 mg/L), ferritin (14,374 ng/mL) and procalcitonin (0.9 ng/mL); low platelet count (=111,000/mm^3^); fibrinogen levels of 230 mg/dL (rapid drop from 430 mg/dL); prothrombin concentration of 45%; D-dimers > 20,000 ng/mL; high values of LDH (2990 U/L) and AST > ALT (1074 > 811 U/L, respectively); total bilirubin (1.39 mg/dL); NT-proBNP (13,422 ng/mL); renal and cardiac markers (troponin, CK-MB) with normal values. The ECG showed sinus tachycardia (127 bpm) and an inverted T-wave in leads III, aVF, V1–V3 with no ST deviation. Dynamic ECGs were not suggestive of acute coronary syndrome. 

Septic shock with disseminated intravascular coagulation or a high-risk (massive) pulmonary thrombosis (PT) were considered. The SOFA score was 12 points. The patient received i.v. fluids, vasopressors and was switched from LMWH to unfractionated heparin (UFH). Extensive microbiological samples were collected, and broad-spectrum i.v. antibiotic and antifungal therapy were started. Remdesivir and Anakinra were stopped because of acute ischemic hepatitis. Echocardiography revealed paradoxical septal motion, right ventricular dilation with severe dysfunction, mild tricuspid regurgitation and a dilated inferior vena cava. A CT pulmonary angiogram (CTPA) confirmed the massive PT (Figure 1, left) and described a progression of COVID-19 pneumonia (60% of lung parenchyma) (Figure 1, right). No clinical or ultrasound imaging signs of deep venous thrombosis were found.

Thrombolysis with alteplase was performed in a timely efficient manner, in less than 48 h after clinical suspicion of the diagnosis. The antimicrobial drugs were stopped. The patient’s haemodynamics stabilised, with no need for further cardiocirculatory support, and inflammatory markers markedly regressed. In the following days, an initial improvement in respiratory function was recorded as the patient was switched to high-flow oxygen therapy, but COVID-19-associated lung lesions deteriorated progressively. The repeated CTPA at 11 days showed a progression of the lung lesions compatible with COVID-19 (80% of the parenchyma) and the persistence of thrombus images, especially on the right side. In the next few days, she required orotracheal intubation and invasive mechanical ventilation and died by multiple organ dysfunction syndrome. 

The diagnosis was a high-risk (massive) PT despite therapeutic doses of anticoagulation in a patient with COVID-19 pneumonia and no other known risk factors for thrombotic disease, based on the classical Wells score. High-risk PT is defined as haemodynamic instability, combined with the confirmation of a CTPA and/or evidence of right ventricle dysfunction via transthoracic echography, based on the ESC 2019 guidelines [23]. As the diagnosis based on clinical and biological markers was not clear, echocardiography was a useful clinical tool. It showed signs of isolated right ventricular dysfunction, which is extremely unlikely in sepsis, and explained the haemodynamic collapse. It also helped in conducting further exploration via CTPA for a positive diagnosis of PT. In terms of optimal management, thrombolysis in less than 48 h was an initial life-saving intervention.

## 3. Discussions and Literature Review

To further explore the risk factors, site of PT, anticoagulation treatment before the diagnosis of pulmonary thrombotic events and outcomes, we conducted a literature review using free searches in the PubMed database. We searched only papers in English, published since the onset of the COVID-19 pandemic, using the following key terms in various combinations: “COVID-19”, “SARS-CoV-2”, “novel coronavirus” and “pulmonary thrombosis”, “pulmonary embolism”, “thromboembolism” and “anticoagulation”. We restricted our search to papers with full text written in English. We used the term PT to cover both pulmonary embolism and in situ thrombosis. PT risk factors were extracted according to the PADUA Prediction Score, a validated score that can identify patients at risk of PT development [24]. 

We included data from original cohort studies and case reports, which met the inclusion criteria of confirmed SARS-CoV-2 infection and diagnosis of PT despite being under anticoagulation therapy. The severity of the COVID-19 disease was considered as reported by the authors.

We excluded review papers and clinical studies in which data on anticoagulation were missing. 

We analysed data from 13 cohort studies including 4058 patients of which 346 (8.5%) developed PT and nine case reports and case series enrolling 14 patients with PT (Table 1). 

PT developed in patients already on various doses of anticoagulation, especially LMWH, prophylactic in most cases but also intermediate and therapeutic. PT was diagnosed in 35 (47%) of 75 ICU patients despite standard thrombosis prophylaxis with LMWH (nadroparin), while four (3%) of 123 COVID-19 patients admitted to the regular ward were diagnosed with symptomatic PT despite thrombosis prophylaxis [35]. The risk of venous thromboembolism (VTE) in ICU patients was not lower when the standard dose of nadroparin prophylaxis was doubled in the same study [35]. In four cohorts enrolling 1744 patients on prophylactic anticoagulation, 79 (4.5%) developed PT [21,30,31,34]. 

In another study involving 184 ICU COVID-19 pneumonia patients from three Dutch hospitals, the incidence of the composite outcome of arterial and venous thrombotic events despite routine LMWH (nadroparin) prophylaxis was 31%. CTPA and/or ultrasonography confirmed PT in 27% and arterial thrombotic events in 3.7%. PT was the most frequent complication, in 81% of patients. The authors reinforced the recommendation to strictly apply pharmacological thrombosis prophylaxis in all COVID-19 patients admitted to the ICU, increasing the standard prophylaxis towards high-prophylactic doses [21]. 

In a cohort of 26 ICU French patients, six (23%) who developed PT were on a therapeutic anticoagulant dose [33]. 

Among 14 patients from case reports/series, seven (50%) received therapeutic doses, one (7.1%) received intermediate doses and six (42.8%) received prophylactic doses. Eleven patients out of 14 received LMWH and the other three either UFH or anti-vitamin K anticoagulants.

Within the available data, in the majority of cases, a peripheral segmental/subsegmental thrombosis was recorded, but some patients also had an involvement of the main/lobar pulmonary arteries [40,42,45]. High-risk PT was described in only a few patients from different papers [25,28,37,39,40,44,45,46]. The localisation of PT, as a predominant segmental/subsegmental lung microthrombosis, is associated with widespread parenchymal abnormalities [25,29]. In a study involving 198 hospitalised patients, almost all pulmonary thrombotic events (12/13) were developed as segmental/subsegmental thrombi in the ICU patients with severe/critical COVID-19 [35]. The two largest cohorts reporting PT [30,32] did not provide information regarding the site of PT.

We further analysed four cohorts of patients, which reported data on risk factors for thrombosis, outcomes and biological characteristics (CRP, D-dimers) [25,29,31,32]. These included 1445 patients, representing 35% out of the 4058 patients, all of whom had been anticoagulated with LMWH in various doses. There have been 194 (13%) patients who developed PT under anticoagulant therapy. The demographic, clinical, laboratory characteristics and in-hospital mortality of the PT group and non-PT group are displayed in Table 2.

The most significant finding regarding the cardiovascular risks factors was the male sex, with 74.7% in the PT group vs. 58.3% in the non-PT group (*p* < 0.001). We have also noted that among patients with prior stroke, the rate of PT was lower, 2.2% compared to 8.09% for patients without prior stroke. Regarding the rest of the cardiovascular risk factors, no significant differences were found for patients with hypertension, dyslipidemia, diabetes mellitus, ischemic heart disease or chronic heart failure. When compared to classical major vascular risk factors, COVID-19 was independently associated with large vessel occlusion strokes, the risk being 2.4 times higher than among patients without SARS-CoV-2 infection [47].

From the PADUA Prediction Score risk factors, we analysed obesity/body mass index (BMI), history of previous VTE and the presence of an active malignancy. Intriguingly, the presence of an active malignancy was more prevalent in the non-PT group, whereas the history of previous VTE did not have a significant difference between the two studied groups (7.29% vs. 7.21%). 

In other studies, not included in this review due to missing data, the authors reported that the incidence of VTE is high in patients hospitalised with COVID-19, and classical risk factors for thrombosis or other comorbidities are not significantly different between patients with and without pulmonary thrombotic events [48]. Complementary, ICU admission, mechanical ventilation and severe parenchymal abnormalities, but not age and common comorbidities, were associated with PT occurrence [49]. 

Patients with severe COVID-19 were more prevalent in the group of patients with PT (41.9% vs. 19.3%). The overall in-hospital mortality in the PT vs. non-PT groups did not show a significant difference (14.4% vs. 12.7%), although the number of patients with a severe disease form was higher among patients in the PT group. Another study reported a higher in-hospital mortality, up to 30% [25].

Also noteworthy, 13/14 patients described in the case reports/case series presented with a severe form of SARS-CoV-2 infection, out of whom, three had a critical disease form with lesion extensions of >75%. Most of these patients had no other thrombosis risk factors (mean age of 60 years old, three patients with previous VTE, one obese patient and one patient was in the postpartum period), with a cumulated mortality of 28%. 

Pulmonary thrombotic risk factors in patients with COVID-19 are rather inflammation-related (including severe/critical COVID-19) than traditional thromboembolic risk factors [37,38,39,40,43,44,45].

In a case series involving five patients with SARS-CoV-2 infection and associated PT, all of them had severe COVID-19 without other risk factors for thrombosis [38]. In another study, all 30 patients with pulmonary thrombosis had severe and critical COVID-19, and more than half had at least 50% lung involvement in terms of imaging inflammatory lesions [25].

Besides ICU stay, other risk factors associated with VTE in univariable regression analyses were higher levels of inflammatory markers (CRP, D-dimer level), and very few patients had other risk factors for thrombosis (obesity, history of VTE or an active cancer) [35]. Data regarding the inflammation markers showed that patients with PT had higher values in the studies reporting on these differences.

A combination of COVID-19 and traditional risk factors for thrombosis could also lead to the failure of anticoagulation despite therapeutic doses [36]. 

Thrombotic events in COVID-19 are multifactorial, being the results of a complex interplay between inflammation, endotheliopathy, coagulation pathways, platelet dysfunction and fibrinolysis disbalance, and this could explain the failure of anticoagulation [1,50,51,52,53] (Figure 2). 

Therefore, the effect of anticoagulation in preventing pulmonary immunothrombosis is less known in comparison with other kinds of thrombotic events [54]. Because neutrophil extracellular traps (NETs), key molecules in immunothrombosis, are positively charged, unfractionated heparin may be a more effective option for SARS-CoV-2-associated pulmonary thrombosis [55]. Heparin also replaces molecules such as histones from the chromatin backbone of NETs and destroys their stability [55]. Variable responses and possible resistance to anticoagulants could also contribute to thrombosis, especially among critically ill COVID-19 patients [56]. Multiple factors may contribute to heparin resistance in acute inflammatory states, including enhanced heparin clearance, antithrombin deficiency, elevated levels of factor VIII and/or fibrinogen and increased levels of heparin-binding proteins [57]. In a study involving 37 anticoagulated patients admitted to ICU for COVID-19 pneumonia, 75.7% of patients were heparin and LMWH “resistant” [57]. 

For the optimisation of LMWH therapy in preventing thromboembolic disease, monitoring the level of anti-factor Xa (anti-FXa) could be useful. This is already recommended for patients with renal failure and cancer [58]. 

In a retrospective cohort study which included 69 critically ill COVID-19 patients who received different regimens of higher-intensity anticoagulation with enoxaparin, fewer than one-third of the patients achieved anti-FXa levels within the typically acceptable target levels, most values being in subprophylactic and subtherapeutic ranges [56]. In addition, conventional anti-Fxa ranges may not be appropriate in critical COVID-19 patients, particularly those receiving therapeutic anticoagulation, and this could also be explained by the multiple factors known to affect anti-FXa levels, including increased acute-phase reactants and inflammatory markers, which have high values in severe/critical COVID-19 cases [56]. 

Anti-Xa levels and aPTT monitoring may be inaccurate in patients with severe illness and COVID-19, and more data are needed in order to optimise the overall management of these patients [59].

## 4. Conclusions

Pulmonary thrombosis should be considered as a differential diagnosis in patients hospitalised with severe and critical COVID-19 when there is a deterioration of respiratory function and an elevation of D-dimers. Thrombotic events associated with severe/critical COVID-19 are not rare in patients receiving anticoagulant therapy, especially in severe and critical disease, where there is a complex interplay between inflammation and coagulation pathways. 

Excessive inflammation seems to be the main driver, as well-known classical prothrombotic risk factors are absent in most cases. Peripheral, segmental/subsegmental thrombi predominate, but high-risk/massive lung thrombosis is also possible. The type, duration of anticoagulation and absence of risk factors for thrombotic disease should not be a barrier to rule out this important, life-threatening diagnosis. The therapeutical approach should take into consideration the mechanisms of pulmonary thrombosis in COVID-19 that are strongly inflammatory-driven.

Nevertheless, the results of this review are limited by the heterogeneity of the population included in the analysed studies in terms of disease severity, imaging and the specific data on anticoagulation. Further prospective studies addressing these issues are necessary.

## Figures and Tables

**Figure 1 viruses-15-01535-f001:**
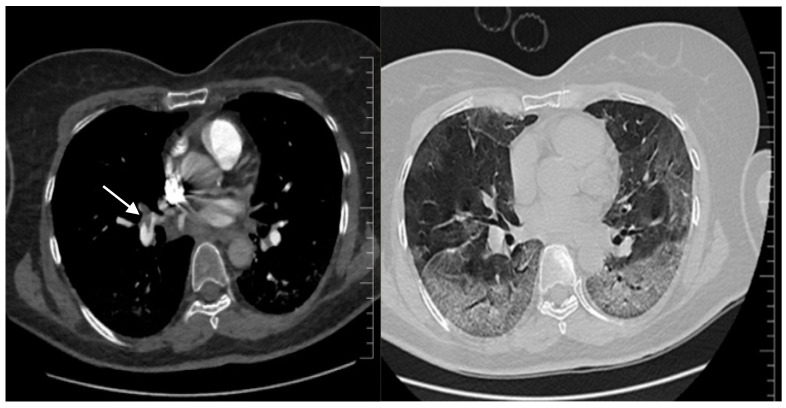
Second chest CT imaging, CT pulmonary angiography: on the left—main pulmonary right artery massive pulmonary embolism (arrow) and extending to the left pulmonary arteries; on the right—severe COVID-19 pneumonia, with lesions involving 60% of the lung parenchyma.

**Figure 2 viruses-15-01535-f002:**
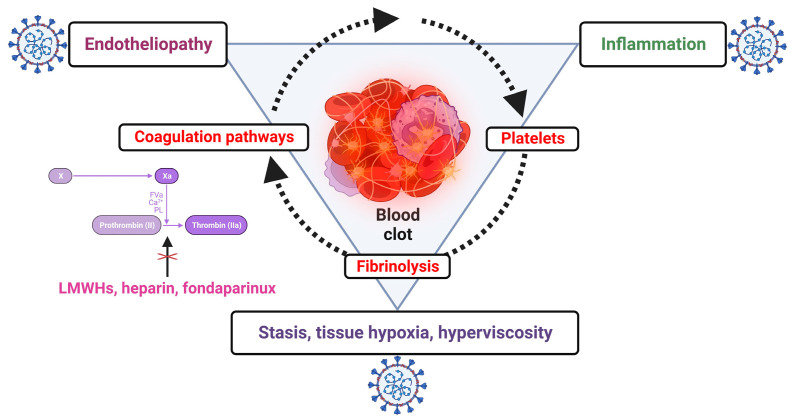
The complex interplay between coagulation, inflammation and other factors explaining the failure of anticoagulation alone for preventing pulmonary thrombotic events in SARS-CoV-2 infection; low-molecular-weight heparins (LMWHs), heparin and fondaparinux interfere only with coagulation pathways (created with Biorender.com).

**Table 1 viruses-15-01535-t001:** Summary of published cohorts and cases with pulmonary thrombosis in anticoagulated patients.

First Author, Reference	Study Type, Patients with PT/Cohort, N	Anticoagulant	Anticoagulation Regimen	Pulmonary Thrombosis Localisation
Niculae, 2022 [25]	Cohort study, 30/30	LMWH	Prophylactic, intermediate, therapeutic	Main/lobar arteries, segmental/subsegmental arteries
Baccellieri, 2021 [26]	Cohort study, 35/200	LMWH	Prophylactic, therapeutic	NA
Bellmunt-Montoya, 2021 [27]	Cohort study, 16/230	LMWH	Prophylactic, intermediate, therapeutic	NA
Di Micco, 2021 [28]	Cohort study, 8/154	LMWH	Prophylactic, intermediate	NA
Jalde, 2021 [29]	C0ohort study, 34/130	LMWH	Prophylactic, intermediate, therapeutic	Segmental/subsegmental thrombosis
Muñoz-Rivas, 2021 [30]	Cohort study, 24/1127	LMWH	Prophylactic	NA
Schiaffino, 2021 [31]	Cohort study, 27/45	LMWH	Prophylactic	Bilateral thrombosis
Fauvel, 2020 [32]	Cohort study, 103/1240	LMWH	Prophylactic, intermediate, therapeutic,	NA
Klok, 2020 [21]	Cohort study, 25/184	LMWH	Prophylactic	Segmental/subsegmental thrombosis
Liitjos, 2020 [33]	Cohort study, 6/18	LMWH	Therapeutic	NA
Lodigiani, 2020 [34]	Cohort study, 3/388	LMWH	Prophylactic	Segmental/subsegmental thrombosis
Middeldorp, 2020 [35]	Cohort study, 13/198	LMWH	Prophylactic, intermediate	Segmental/subsegmental thrombosis, main pulmonary arteries
Poissy, 2020 [36]	Cohort study, 22/106	LMWH	Prophylactic, therapeutic	Bilateral thrombosis, proximal, segmental thrombosis
Belbarbi, 2022 [37]	Case report	LMWH	Prophylactic	Bilateral pulmonary arteries
Di Tano, 2022 [38]	Case series, 5/5	Oral anticoagulation	Prophylactic, therapeutic	Segmental/subsegmental thrombosis
Aribawa, 2021 [39]	Case report	Heparin	Therapeutic	Multiple thrombosis on pulmonary arteries
Brem, 2021 [40]	Case report, 2/2	LMWH	Prophylactic	Main pulmonary arteries
Kripalani, 2021 [41]	Case report	LMWH	Prophylactic	Lobar inferior artery, segmental/subsegmental thrombosis
Terrigno, 2021 [42]	Case report	Oral anticoagulation	Therapeutic	Right main pulmonary artery
Geerdes-Fenge, 2020 [43]	Case report	LMWH	Prophylactic	Bilateral lobar inferior arteries
Rout, 2020 [44]	Case report	LMWH	Prophylactic	NA
Salam, 2020 [45]	Case report	LMWH	Intermediate	Bilateral thrombosis, main pulmonary arteries

Legend: PT—pulmonary thrombosis, LMWH—low-molecular-weight heparin, NA—not available.

**Table 2 viruses-15-01535-t002:** Demographic, clinical, laboratory characteristics and mortality of patients in PT and non-PT groups in 4 analysed cohort studies with anticoagulated patients.

Article	Niculae, 2022 [25] *n* = 30		Jalde, 2021 [29] *n* = 130		Schiaffino, 2021 [31] *n* = 45		Fauvel, 2020 [32] *n* = 1240		Total *n* = 1445		*p*-Value	OR (95% CI)
	PT *n* (%)	Non-PT *n* (%)	PT *N* (%)	Non-PT *n* (%)	PT *n* (%)	Non-PT *n* (%)	PT *n* (%)	Non-PT *n* (%)	PT *N* (%)	Non-PT *n* (%)		
	30 (100)	0	34 (26.2)	96 (73.8)	27 (60)	18 (40)	103 (8.3)	1137 (91.7)	194 (13.4)	1251 (86.6)		
Cardiovascular risk factors												
Age, (median/mean)	62 (54–74)	NA	58 (51–64)	58 (50–65)	70	62	63 ± 16	64 ± 17	NA	NA	-	-
Male sex, *n* = 1445	25 (83.3)	NA	27 (79.4)	68 (70.8)	20 (74)	14 (77.7)	73 (70.9)	648 (57)	145 (74.7)	730 (58.3)	<0.001	2.1 (1.4, 2.9)
Hypertension, *n* = 1445	15 (50)	NA	14 (44)	34 (36)	13 (48.1)	10 (55.5)	44 (42.7)	515 (45.7)	86 (44.32)	559 (44.68)	0.92	0.98 (0.7, 1.3)
Dyslipidemia, *n* = 1400	7 (23.3)	NA	6 (18)	10 (11)	NA	NA	22 (21.4)	294 (26)	35 (20.95)	304 (24.65)	0.29	0.81 (0.5, 1.2)
Diabetes mellitus, *n* = 1400	9 (30)	NA	4 (12)	18 (19)	NA	NA	19 (18.4)	249 (22)	32 (19.16)	267 (21.65)	0.46	0.85 (0.5, 1.2)
Prior stroke, *n* = 1270	1 (3.3)	NA	NA	NA	NA	NA	2 (1.9)	92 (8.2)	3 (2.25)	92 (8.09)	<0.01	0.26 (0.08, 0.8)
IHD, *n* = 1400	2 (6.6)	NA	2 (6)	7 (8)	NA	NA	9 (8.7)	124 (10.9)	13 (7.78)	131 (10.62)	0.25	0.71 (0.3, 1.2)
Smoking, *n* = 1240	NA	NA	NA	NA	NA	NA	9 (8.9)	172 (15.4)	9 (8.73)	172 (15.12)	0.06	0.53 (0.2, 1)
CHF, *n* = 1400	2 (6.6)	NA	0	7 (7.2)	NA	NA	12 (11.8)	105 (9.3)	14 (8.38)	112 (9.08)	0.78	0.91 (0.5, 1.6)
Other classical risk factors for thrombosis												
BMI/Obesity, (median/mean)	8 (26.6)	NA	27 (23–29)	28 (24–31)	26	28	27.3 ± 5.6	28.2 ± 6.3	NA	NA	-	-
Active malignancy, *n* = 1445	1 (3.3)	NA	4 (12)	14 (15)	2 (7.4)	2 (11.1)	8 (7.8)	159 (14)	15 (7.73)	175 (13.98)	0.01	0.51 (0.2, 0.8)
Prior VTE, *n* = 1370	NA	NA	0	1 (1)	NA	NA	10 (9.7)	88 (7.4)	10 (7.29)	89 (7.21)	0.94	1.01 (0.5, 1.9)
Biological characteristics												
CRP, mg/L (median/mean)	40.4 ±41.4	NA	295 (250–341)	214 (187–241)	42	2	114 ± 95	89 ± 75	NA	NA	-	-
D-dimers, mg/L (median/mean)	7.4 ± 6.9	NA	8.2 (5.8–11.8)	2.1 (1.6–2.7)	0.7	0.5	3.5 ± 4.3	1.3 ± 4.1	NA	NA	-	-
COVID-19 severity and in-hospital mortality												
Severe COVID-19 disease *, *n* = 1445	22 (73.3)	0	22 (64.7)	53 (55.2)	NA	NA	26 (25.2)	185 (16.2)	70 (41.9)	238 (19.3)	<0.001	3.01 (2.1, 4.2)
Deceased, *n* = 1445	9 (30)	0	8 (23.5)	14 (15)	2 (7.5)	1 (5.5)	9 (8.7)	142 (12.5)	28 (14.4)	157 (12.5)	0.46	1.17 (0.7, 1.7)

* In the analysed cohorts, a severe form of COVID-19 disease is defined as parenchymal abnormalities on a CT affecting >50% of the lungs. Legend: PT—pulmonary thrombosis, NA—not available, IHD—ischemic heart disease, CHF—chronic heart failure, BMI—body mass index, VTE—venous thromboembolism, CRP—C-reactive protein. Descriptive data were expressed as frequencies (%) for categorical data; quantitative variables were expressed as the mean  ±  standard deviation (SD) or the median (interquartile range, IQR). Statistical significance was set at *p* < 0.05. Data were analysed using OpenEpi, Version 3.01.

## Data Availability

Not applicable.

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
