# Peer review of "Pulmonary Thrombosis despite Therapeutic Anticoagulation in COVID-19 Pneumonia: A Case Report and Literature Review"

_viruses, 2023, doi:10.3390/v15071535_

Round 1

Reviewer 1 Report

Comments and Suggestions for Authors

The review by Niculae, C-M. et al. is very well written and it is very clear to see that the authors have done a thorough investigation into pulmonary thrombosis across different cohorts in the context of anti-coagulation treatment as well as other factors. I only have a few minor comments:

For the sentence "Coronavirus disease 2019 (COVID-19), is a complex respiratory and systemic disease, characterised by severe inflammation which leads to a prothrombotic state and pulmonary lesions.", for most infected with SARS-CoV-2, COVID-19 is a very mild disease and not always characterised by severe inflammation and definitely doesn't always lead to a prothrombotic state. There are multiple disease states for COVID-19 and it should be stated that the disease can progress to severe inflammation, which can lead a prothrombotic state.

For the sentence "The rate of thrombotic complications in COVID-19 patients is high and associated with the risk of unfavourable outcomes, therefore, adequate screening, diagnostic and antithrombotic strategies are essential.", there should be a reference for this statement. It is also unclear what a high rate of thrombotic complications is. Is this high relative to other respiratory diseases? Or is it overall just a high rate. This should be slightly clearer.

For "a positive SARS-CoV-2 RT-PCR nasopharyngeal swab test.", I don't think the word "test" needs to be there.

For "She was unvaccinated against SARS-CoV-2.", It is more correct to say unvaccinated against COVID-19 as the vaccines were not designed to prevent SARS-CoV-2 infection and, instead, designed to prevent progression to severe COVID-19. 

Reviewer 2 Report

Comments and Suggestions for Authors

Dear Author!

Thank you so much for the interesting data presented. The clinical case is interesting and reflects the problem very well. I appreciate the review that you conducted. It is informative and helps to understand magnitude of the pulmonary artery thrombosis as well as existing lack of knowledge regarding its pathogenesis, prophylaxis and effective management.

I have just two minor remarks that should be adressed.

1. Please, report when the clinical case had happened, at least month and year 

2. As I understand from Table 2 you performed pooling of the data and analyzed it. This has to be stated clearly with explaining what and which way was performed. Please, specify also which statistical instruments were used. 

Comments on the Quality of English Language

None
